# Design and Synthesis of Novel Imidazole Derivatives Possessing Triazole Pharmacophore with Potent Anticancer Activity, and In Silico ADMET with GSK-3β Molecular Docking Investigations

**DOI:** 10.3390/ijms22031162

**Published:** 2021-01-25

**Authors:** Fawzia Al-blewi, Salma Akram Shaikh, Arshi Naqvi, Faizah Aljohani, Mohamed Reda Aouad, Saleh Ihmaid, Nadjet Rezki

**Affiliations:** 1Department of Chemistry, College of Science, Taibah University, Al-Madinah Al-Munawarah 30002, Saudi Arabia; shfaaf5@gmail.com (S.A.S.); arshi_84@yahoo.com (A.N.); m.sfm@hotmail.com (F.A.); aouadmohamedreda@yahoo.fr (M.R.A.); 2Pharmacognosy and Pharmaceutical Chemistry Department, College of Pharmacy, Taibah University, Al-Madinah Al-Munawarah 30002, Saudi Arabia; saleh_ihmaid@yahoo.com.au

**Keywords:** 1,2,3-triazole, imidazole, click synthesis, anticancer activity, docking study

## Abstract

A library of novel imidazole-1,2,3-triazole hybrids were designed and synthesized based on the hybrid pharmacophore approach. Therefore, copper(I)catalyzed click reaction of thiopropargylated-imidazole 2 with several organoazides yielded two sets of imidazole-1,2,3-triazole hybrids carrying different un/functionalized alkyl/aryl side chains **4a**–**k** and **6a**–**e**. After full spectroscopic characterization using different spectral techniques (IR, ^1^H, ^13^C NMR) and elemental analyses, the resulted adducts were screened for their anticancer activity against four cancer cell lines (Caco-2, HCT-116, HeLa, and MCF-7) by the MTT assay and showed significant activity. In-silico molecular docking study was also investigated on one of the prominent cancer target receptors, i.e., glycogen synthase kinase-3β (GSK-3β), revealing a good binding interaction with our potent compound, **4k** and was in agreement with the in vitro cytotoxic results. In addition, the ADMET profile was assessed for these novel derivatives to get an insight on their pharmacokinetic/dynamic attributes. Finally, this research design and synthesis offered click chemistry products with interesting biological motifs mainly 1,2,3 triazoles linked to phenyl imidazole as promising candidates for further investigation as anticancer drugs.

## 1. Introduction

Despite several anticancer drugs being available in the market hitherto, many impediments of current anticancer remedies have implored researchers to keep looking for new candidates [1,2,3]. However, heterocycles are well documented as fascinating skeletons when constructing molecules that will interact with the targets and disrupt the biological pathways associated with cancer progression. In addition, the relative ease in designing these ring systems with several substituents enables them to cover a wide range of chemical space and further making them as significant starting points for anticancer drug design and discovery [4,5].

Being a group of highly diversified structures, nitrogen rich heterocycles and their fused systems are widely incorporated into the structure of various pharmacologically active agents and synthetic drugs [6]. Among these bioactive heterocycles, imidazole derivatives [7], particularly substituted thioimidazoles are known in medicinal chemistry as anti-obesity [8], antitubercular [9], antidiabetic [10], anticancer [11,12,13], antimicrobial [14,15], and antioxidant agents [16].

1,2,3-triazoles have also marked their position as a significant pharmacophore from nitrogen rich heterocyclic compounds with spectacular therapeutic potential [17,18]. Rufinamide (anticonvulsant), TSAO (anti-HIV), cefatrizine (antibiotic), tazobactam (antibacterial), and CAI (anticancer) are some of the FDA approved drugs bearing 1,2,3-triazole moiety [19,20,21,22].

Lately, it has been observed that among the combinatorial libraries of derivatized heterocycles, the most active ones had a bi-heterocyclic structure, which provides better binding opportunities in comparison to mono-heterocycles [23,24,25]. Consequently, we envisioned the molecular coupling of thioimidazole derivative with the 1,2,3-triazole core through a flexible spacer. The resulting molecular hybrids could deliver better interactions with biological targets by means of their synergistic effects [26].

Physicochemical, pharmacokinetic, and pharmacodynamics features, over the past few decades, have emanated as one of the crucial phases in drug discovery. This “pharmaceutical profiling” strategy furnishes a variety of suitable “drug-like” traits that can be developed into a structure-property association. Jointly, this information is utilized to escort chemical synthesis and nominate the best candidates for further drug advancement. Drug discovery and research organizations have set up a capability for the structured in vitro computation of properties which are pivotal to the understanding of absorption, distribution, metabolism, excretion, and toxicity (ADMET) behavior in vivo. In silico ADMET forecasting is expected to minimize the probability of late-stage attrition of drug development procedure and to optimize screening and trials by gazing at the promising drug candidates only. Molecular docking is a computational simulation strategy of an aspirant ligand binding to a macromolecule (receptor) and foretells the favorable orientation of binding one compound to the second one to make a stable complex. In silico molecular docking is employed to forecast the binding affinity and activity of the small molecule to their target receptors by availing scoring functions. Hence, the docking approach plays an influential role in the rational design of drugs in the drug development process. The synthesized ligand **4k** was appraised by the in silico docking analysis for procuring an additional comprehension of the binding pattern with the glycogen synthase kinase-3 (GSK-3). Glycogen synthase kinase-3 (GSK-3), a serine/threonine protein kinase, was initially described as a key enzyme involved in glycogen metabolism [27,28]. Recent reports have suggested that GSK-3β is a positive regulator of cancer cell proliferation and survival [29,30,31,32], thus, providing further support for GSK-3β as a therapeutic target in cancer.

Considering the facts mentioned above, and pursuing our ongoing interest in the design of bioactive 1,2,3-triazoles molecular conjugates [33,34,35,36,37,38,39,40,41,42,43], we describe herein the design and synthesis of focused 1,4-disubstituted 1,2,3-triazoles, linked to the imidazole ring via regioselective copper catalyzed-1,3-dipolar cycloaddition reaction of *S*-propargylimidazole and some selected un/functionalized alkyl and aryl azides. The newly synthesized adducts were subjected to in vitro anticancer evaluation utilizing an MTT colorimetric assay against four different human cancer cell lines (Caco-2, HCT-116, HeLa, and MCF-7). The physicochemical and ADMET properties were also forecasted. Further mechanistic computational analysis was done by molecular docking screening of the synthesized click product **4k** using GSK-3β as the target receptor. A schematic diagram for the study work is given in Figure 1.

## 2. Results and Discussion

### 2.1. Chemistry

Our strategy for designing two sets of novel bi-heterocyclic based on imidazole-1,2,3-triazole combined systems was driven by employing a substituted imidazole, inserting thiomethylene bridge as a flexible linker between imidazole and triazole moieties, along with derivatizing triazole at the N-1 position with flexible (Scheme 2) and rigid (Scheme 3) groups to enhance the solubility and/or bioavailability of the resulting products, which is incredibly essential for crossing the cell membrane and reaching the target. Initially, 1,4,5-triphenylimidazole-2-thione (**1**) upon reaction with propargyl bromide in refluxing the methanolic solution of sodium methoxide for 2 h, yielded the desired thioimidazole-based alkyne **2** in 90% yield, Scheme 1.

The formation of the thiopropargylated imidazole **2** has been approved based on its spectral analysis. The sharp band observed at 3300 cm^−1^ in the IR spectrum indicated the presence of the acetylenic hydrogen (≡C–H), while the (C≡C) group appeared also as a sharp band at 2140 cm^−1^. Additionally, its ^1^H NMR revealed the disappearance of the S**H** proton of its starting material **1** and the appearance of a new significant singlet in the aliphatic region at δ_H_ 3.25 and 3.99 ppm assigning the sp-C**H** and SC**H_2_** protons, respectively, guaranteed the incorporation of a propargyl side chain. The aromatic protons of the phenyl rings have been recorded at their usual chemical shifts (δ_H_ 7.18–7.46 ppm). Moreover, the ^13^C NMR spectrum exhibited the diagnostic propargyl carbon signals (S**C**H_2_ and **C**≡**C**) at δ_C_ 21.54, 74.81, and 80.47 ppm, respectively. The signals observed at δ_C_ 126.72–141.39 ppm were assigned to the aromatic and **C**=N carbons.

The synthesized alkyne derivative **2** was then coupled with a rich variety of organic azides **3a–k** and **5a–e** via the 1,3-dipolar cycloaddition reaction catalyzed by copper sulphate and sodium ascorbate in a mixture of DMSO:H_2_O (1:1) at room temperature to afford the targeted imidazole-1,2,3-triazole hybrids carrying un-/functionalized alkyl and/or phenyl acetamide side chain **4a**–**k** (86–93%) and **6a**–**e** (85–86%) and possessing rigid functionalized aromatic units; as described in Scheme 2 and Scheme 3, respectively.

The spectroscopic data including IR, ^1^H NMR, ^13^C NMR, and elemental analyses of all newly designed hybrids **4a**–**k** and **6a**–**e** approved the success of the click reaction. By taking the adduct 1,2,3-triazole **4e** as a model, its IR spectrum disclosed the absence of absorption bands related to the C≡C and ≡C-H groups and the presence of characteristic bands at 1725, 2889, and 2967 cm^−1^ related to the C=O group and aliphatic hydrogens, respectively. Additionally, the ^1^H NMR spectrum exhibited a diagnostic singlet at δ_H_ 8.04 ppm attributed to the C**H**-1,2,3-triazole ring. The spectrum also showed characteristic triplet and quartet at δ_H_ 1.20 and 4.14-4.19 ppm belonging to the ester functionality. While the SC**H_2_** and NC**H_2_** protons resonated as two singlets at δ_H_ 4.50 and 5.39 ppm, respectively. Furthermore, the ^13^C NMR analysis supported the proposed structure **4e** by the absence of the alkyne carbons of its precursor alkyne **2** and the presence of new signals at δ_C_ 14.43 and 61.34 ppm attributed to the ester side chain carbons. The presence of the ester functionality has been also confirmed by the appearance of a downfield signal at δ_C_ 167.74 ppm assigned to the ester carbonyl carbon.

### 2.2. Biological Screening

#### 2.2.1. In Vitro Anticancer Study

The synthesized imidazole-1,2,3-triazole hybrids were screened for their anticancer activity against three different types of cancers namely colon, cervical, and breast cancers using doxorubicin as the reference drug. Cancer cell lines included Caco2 and HCT116 (human colon carcinoma), HeLa (human cervical carcinoma), and MCF-7 (human breast adenocarcinoma). The results were reported as half maximal inhibitory concentration (IC_50_) caused by the tested candidates (Table 1).

Initial screening revealed that most of the synthesized triazoles showed very good to good activity, i.e., their IC_50_ lies in the range of 4.67–20.69 µM (Caco-2), 4.80–30.41 µM (HeLa), and 0.38–27.29 µM (MCF-7), except **2**, **4d**, **4e, 6c,** and **6d**. Interestingly, a preliminary structure–activity relationship analysis urged the low activity of the acetylated **4e, 6c,** and **6d** derivatives due to the presence of acetyl functionality in their structures in the form of ketone or ester composition. The *S*-propargylimidazole 2 (high IC_50_) have not displayed any significant anticancer activity against all the used cell lines, confirming the benefit of the molecular hybridization. Among the newly synthesized click adducts, the triazole **4k** was found to be one of the most potent derivatives against MCF-7 cell lines with IC_50_ of 0.38 µM and demonstrated similar potency to the standard drug, doxorubicin. This could be presumably due to the presence of a lipophilic long alkyl side chain appended to the 1,2,3-triazole ring in compound **4k.** Moreover, the imidazole triazole conjugates, **4k** and **6e** bearing aromatic carboxylic group in their structures, displayed better potency amongst the synthesized compounds against the Caco-2 cell line with IC_50_ of 4.67 ± 0.11 µM and 5.22 ± 0.20 µM, respectively, significantly similar to the standard doxorubicin (IC_50_ = 5.17 ± 0.25 µM). The replacement of aromatic carboxylic group by the fluorinated phenyl group decreased the anticancer potency against the Caco-2 cell line, compound **4i** (IC_50_ = 6.31 ± 0.17 µM) and compound **4g** (IC_50_ = 8.45 ± 0.18 µM). Whereas, triazoles **4g** and **4h** having electron withdrawing groups (Cl, NO_2_) showed a moderate anticancer activity ranging from 12.31 ± 0.22 µM to 15.78 ± 0.31 µM. These results suggested that, the designed triazoles are promising candidates for the future anticancer molecule discovery and research.

#### 2.2.2. In Silico ADMET Analysis

In silico predictions of the physicochemical descriptors, drug likeness or ADMET (adsorption, distribution, metabolism, excretion, and toxicity) properties, have increased the possibility of detecting new lead compounds in a much shorter time span as compared to the conventional/traditional procedures. In silico studies were conducted to confirm the precision of in vitro biological outcomes. Various physicochemical traits, i.e., the presence of a particular class of atoms or bonds, lipophilicity, molar refractivity, and solubility in water and TPSA (topological polar surface area) were computed. These physicochemical properties are given in Table 2. The forecasted physicochemical characteristics are in agreement with the exercised criteria and are presumed to possess a good bioavailability score as all the compounds have TPSA ≤ 140 Å^2^ except compound **4h**.

On the other hand, ADME predictions are documented in Table 3, revealing that all the compounds have low gastrointestinal absorptions (GI) except compound **4e**. All the compounds displayed no blood brain barrier (BBB) permeation. Few of the tested compounds are P-gp (p-glycoprotein) inhibitors, i.e., **2**, **4a–4c,** and **4e**. The BOILED-Egg diagram which is a pooled built-in graphical classification model for the prediction of BBB permeations, passive human gastrointestinal absorption (HIA), and P-gp substrates, is shown in Figure 2. In terms of metabolism, all the synthesized compounds were tested for inhibition of the Cytochrome P450 isomers, i.e., CYP1A2, CYP2C19, CYP2C9, CYP2D6, and CYP3A4. Most of the compounds are non-inhibitors of CYP1A2 and CYP2D6 with the exception of **4b**, as it is a CYP1A2 inhibitor and **4a** which also inhibits both CYP1A2 and CYP2D6. Most of the tested molecules are CYP3A4 inhibitors. The tested compounds emerged to be non-inhibitors of CYP2D6. The skin permeability coefficient (log Kp; with Kp in cm/s) values display low permeability through the skin for most of the compounds except, compound **2**, **4a–4c** which displayed good skin permeability.

Five different rule-based sieves, with diverse ranges of characteristics inside of which the drug candidate is defined as drug-like were appraised, i.e., Lipinski [44], Ghose [45], Veber [46], Egan [47], and Muegge [48] rules. Drug-like predictions with a number of violations to the above-mentioned rules, their bioavailability scores, and toxicity profiles are documented in Table 4. All the tested compounds violated the Ghose and Muegge rules. The Lipinski and Egan rules are violated by most of the compounds except compound **4e**. Most of the compounds follow the Veber rule with few exceptions. Our results indicated that the tested compounds were in good agreement in terms of the bioavailability score ranging from 0.17–0.55. Compounds **4e**, **6a**, **6d,** and **6e** are predicted to be non-toxic in nature by AMES, i.e., they are not mutagenic. All the synthesized compounds are forecasted as non-carcinogens. Doxorubicin was predicted as AMES toxic, mutagenic, and non-carcinogenic in nature. 

#### 2.2.3. Molecular Docking Simulations

Glycogen synthase kinase-3 beta (GSK-3β) is a promising target with an overexpressed oncogene in different breast cancers [49], so we evaluated the binding modes of our potent compound **4k** as a selective MCF-7 cytotoxic agent against the GSK-3β ATP binding pocket and confirmed the overall prognostic dependency of breast cancer patients on the GSK-3β inhibition effect. We docked the **4k** test ligand on our target protein, i.e., GSK-3β (PDB ID: 1UV5), in order to analyze the binding pattern and affinity using the software MOE. Docking poses of the bound and tested compound **4k** are depicted in Figure 3A–C. The compound **4k** displayed an excellent binding interaction and affinity, i.e., −9.8 kcal/mol toward GSK-3β target compared to the reference drug; 6-Bromoindirubin-3′-oxime with −8.8 kcal/mol. The interaction analysis of the bound ligand; 6-bromoindirubin-3′-oxime, exhibited that the cyclic nitrogen of the pyrrole ring donates a hydrogen bond to the peptide carbonyl oxygen of Val135, while the cyclic nitrogen of the lactam ring donates a hydrogen bond to the carbonyl oxygen of Asp133, and the lactam carbonyl oxygen accepts a hydrogen bond from the backbone amide of Val135 and Tyr134. The molecular docking studies revealed that the compound **4k** emerged to be a promising drug candidate; on the account of its lowest binding energy, i.e., −9.8 kcal/mol with the active site residues of the target receptor GSK-3β and this might be one of the reasons for its good in vitro anticancer activity. The putative interaction of compound **4k** with GSK3 suggested that the terminal aromatic ring might display two hydrogen bond interactions through the COOH substituent with the backbone carbonyl group of Asp133 and the backbone amide of Val135, respectively. Moreover, another essential linker hydrogen bond through the carbonyl of amide with conserved Cys199 stabilized the compound in the pocket. In addition, the fit of the aromatic systems for the whole compound occurs through a group of aromatic stacking interactions with corresponding residues; Ile62, Tyr134, Asn64 for triazole ring, Val70, and Leu188 that might support selective binding. The 2D and 3D analysis of the target compound **4k** mapped to the 6-Bromoindirubin-3′-oxime as a reference is shown in Figure 3. Taken together, these observations might rationalize the observed cytotoxic activity of **4k** through the inhibitory effect of overexpressed GSK-3β within the cancer cell lines.

## 3. Experimental Section

The ^1^H NMR spectra were recorded using an Advance Bruker NMR spectrometer (Bruker, Switzerland) at 400–600 MHz, while ^13^C NMR spectra were recorded on the same instrument at 100–150 MHz using tetramethylsilane (TMS) (d, ppm) as the internal standard. The EI mass spectra were measured with a Finnigan MAT 95XL spectrometer (Finnigan, Germany). Sonochemical reactions were performed in a Kunshan KQ-250B ultrasound cleaner (50 kHz, 240 W, China).

All the solvents and reagents used in this work were of the highest quality of analytical reagent grade and purchased from Sigma-Aldrich, USA, and were used without further purification. All the reactions were monitored by thin layer chromatography (TLC), using UV fluorescent Silica gel type Merck 60 F254 plates. The spots were visualized using a UV lamp (254 nm). The melting points of the synthesized products were measured using a Stuart Scientific SMP1 (Stuart, UK). The functional groups were identified using a SHIMADZU FTIR-Affinity-1S spectrometer in the range of 400–4000 cm^−1^ using a Perkin-Elmer 1430 series FT-IR spectrometer (Perkin-Elmer, USA) as potassium bromide pellets. All the synthesized compounds were fully characterized by ^1^H, ^13^C NMR, and elemental analysis. The ^1^H NMR (400 MHz) and ^13^C NMR (100 MHz) spectra were investigated with an Advance Bruker NMR spectrometer (Bruker, Switzerland) (400 MHz) with TMS as an internal standard to calibrate the chemical shifts (δ) reported in ppm. Elemental analyses were performed using a GmbH, Vario EL III, Elementar Analyzer (HEKAtech GmbH, Germany).

### 3.1. Synthesis of the Title Compounds

The azides **3a–k** and **5a–e** used in this study were prepared according to the reported procedures [50,51,52]. The procedures for the synthesis of the reported compounds are described below. 

#### 3.1.1. Synthesis and Characterization of 1,4,5-triphenyl-2-(prop-2-yn-1-ylthio)-1H-imidazole (**2**)

A solution of imidazole **1** (12 mmol) in a methanolic solution (30 mL) of sodium methoxide (12 mmol) was stirred, then propargyl bromide (12 mmol) was added under stirring. The mixture was heated under reflux for 1 h until the completion of the reaction (as indicated by TLC; hexane—ethyl acetate 2:1). The solvent was removed by evaporation under reduced pressure; the solid formed was collected by filtration, washed with water, dried, and recrystallized from ethanol to give the desired imidazole-based alkyne **2** as colorless crystals in 90% yield, mp: 103–105 °C. IR (*υ*, cm^−1^): 1590 (C=C), 1620 (C=N), 2140 (C≡C), 2888, 2894 (C-H Al), 3050 (C-H Ar), 3300 (≡CH). ^1^H NMR (400 MHz, DMSO-*d_6_*): δ_H_ = 3.25 (s, 1H, ≡C**H**), 3.99 (s, 2H, SC**H_2_**), 7.18–7.28 (m, 10H, Ar-**H**), 7.39–7.46 (m, 5H, Ar-**H**) ppm. ^13^C NMR (100 MHz, DMSO-*d_6_*): δ_C_ = 21.54 (S**C**H_2_); 74.81 (C≡**C**H); 80.47 (**C**≡CH); 126.72, 127.09, 128.68, 128.73, 128.94, 129.05, 129.53, 129.65, 130.40, 131.17, 131.89, 134.44, 135.59, 138.08, 141.39 (Ar-**C**, **C**=N) ppm. Calculated for C_24_H_18_N_2_S: C, 78.66; H, 4.95; N, 7.64. Found: C, 78.59; H, 4.89; N, 7.82.

#### 3.1.2. General Procedure for the Synthesis of 1,4-disubstituted 1,2,3-triazoles Bearing Imidazole Moiety **4a**–**k** and **6a**–**e**

A solution of copper sulfate (0.10 g) and sodium ascorbate (0.15 g) in water (10 mL) was added dropwise to a solution of alkyne **2** (1 mmol) in DMSO (10 mL) under stirring. Then, the appropriate azide **3a–k** and/or **5a–e** (1 mmol) was added. The stirring was continued for 6-10 h at room temperature. The reaction was monitored via TLC (hexane-ethyl acetate 2:1), and after the completion of the reaction, the mixture was poured onto iced-water. The precipitate thus formed was collected by filtration, washed with a saturated solution of ammonium chloride, and recrystallized from ethanol/DMF to give the targeted 1,2,3-triazoles **4a**–**k** and **6a**–**e**.

##### 4-(((1,4,5-Triphenyl-1H-imidazol-2-yl)thio)methyl)-1-undecyl-1H-1,2,3-triazole (**4a**)

This compound was obtained as a yellow solid in 90% yield, mp: 99–101 °C. IR (*υ*, cm^−1^): 1525 (C=C), 1610 (C=N), 2910-2880 (CH-Al), 3030 (CH-Ar). ^1^H NMR (400 MHz, DMSO-*d_6_*): δ_H_ = 0.87 (t, 3H, *J* = 4.0 Hz, C**H**_3_), 1.19-1.28 (m, 16H, 8×C**H**_2_), 1.77 (bs, 2H, NCH_2_C**H**_2_), 4.27 (t, 2H, *J* = 4.0 Hz, NC**H**_2_), 4.46 (s, 2H, SC**H**_2_), 7.18-7.29 (m, 10H, Ar-**H**), 7.33-7.46 (m, 5H, Ar-**H**), 8.08 (s, 1H, C**H**-1,2,3-triazole) ppm. ^13^C NMR (100 MHz, DMSO-*d_6_*): δ_C_ = 14.45 (**C**H_3_); 26.15, 28.66, 29.22, 29.30, 29.36, 29.38, 30.14 (**C**H_2_); 27.53 (S**C**H_2_); 54.87 (N**C**H_2_); 125.42, 126.62, 126.77, 127.06, 128.65, 128.68, 128.86, 129.03, 129.45, 129.63, 130.38, 131.13, 131.64, 134.57, 135.59, 142.39, 142.46 (Ar-**C**, **C**=N) ppm. Calculated for C_27_H_25_N_5_S: C, 74.56; H, 7.33; N, 12.42. Found: C, 74.78; H, 7.45; N, 12.66.

##### 4-(((1,4,5-Triphenyl-1H-imidazol-2-yl)thio)methyl)-1-hexadecyl-1H-1,2,3-triazole (**4b**)

This compound was obtained as a light-yellow solid in 88% yield, mp: 90–91 °C. IR (*υ*, cm^−1^): 1530 (C=C), 1620 (C=N), 2955 (C-H Al), 3050 (C-H Ar). ^1^H NMR (400 MHz, DMSO-*d_6_*): δ_H_ = 0.88 (t, 3H, *J* = 4.0 Hz, C**H**_3_), 1.16-1.26 (m, 26H, 13×C**H**_2_), 1.77 (t, 2H, *J* = 4.0 Hz, NCH_2_C**H**_2_), 4.29 (s, 2H, NC**H**_2_), 4.45 (s, 2H, SC**H**_2_), 7.20-7.28 (m, 10H, Ar-**H**), 7.36-7.47 (m, 5H, Ar-**H**), 8.06 (s, 1H, C**H**-1,2,3-triazole) ppm. ^13^C NMR (100 MHz, DMSO-*d_6_*): δ_C_ = 14.46 (**C**H_3_); 22.56, 26.24, 28.79, 29.24, 29.29, 29.39, 29.48, 29.56, 30.13 (**C**H_2_); 27.57 (S**C**H_2_); 54.81 (N**C**H_2_); 125.40, 126.65, 126.76, 127.09, 128.62, 128.65, 128.81, 129.08, 129.48, 129.69, 130.35, 131.10, 131.61, 134.53, 135.55, 142.33, 142.52 (Ar-**C**, **C**=N) ppm. Calculated for C_27_H_25_N_5_S: C, 75.79; H, 8.11; N, 11.05. Found: C, 75.58; H, 8.23; N, 11.29.

##### 4-(((1,4,5-Triphenyl-1H-imidazol-2-yl)thio)methyl)-1-octadecyl-1H-1,2,3-triazole (**4c**)

This compound was obtained as a pale yellow solid in 88% yield, mp: 10–110 °C. IR (*υ*, cm^−1^): 1560 (C=C), 1620 (C=N), 2955 (C-H Al), 3050 (C-H Ar). ^1^H NMR (400 MHz, DMSO-*d_6_*): δ_H_ = 0.85 (t, 3H, *J* = 4.0 Hz, C**H**_3_), 1.14-1.29 (m, 30H, 15×C**H**_2_), 1.74 (t, 2H, *J* = 4.0 Hz, NCH_2_C**H**_2_), 4.26 (s, 2H, NC**H**_2_), 4.49 (s, 2H, SC**H**_2_), 7.21-7.30 (m, 10H, Ar-**H**), 7.35-7.48 (m, 5H, Ar-**H**), 8.10 (s, 1H, C**H**-1,2,3-triazole) ppm. ^13^C NMR (100 MHz, DMSO-*d_6_*): δ_C_ = 14.43 (**C**H_3_); 22.59, 26.27, 26.65, 26.89, 28.72, 29.27, 29.35, 29.48, 29.56, 29.59, 30.15 (**C**H_2_); 27.59 (S**C**H_2_); 54.89 (N**C**H_2_), 125.42, 126.69, 126.79, 127.17, 128.71, 128.69, 128.88, 129.13, 129.42, 129.68, 130.34, 131.13, 131.64, 134.55, 135.58, 142.37, 142.58 (Ar-**C**, **C**=N) ppm. Calculated for C_27_H_25_N_5_S: C, 76.20; H, 8.37; N, 10.58. Found: C, 76.48; H, 8.26; N, 10.77.

##### 4-(((1,4,5-Triphenyl-1H-imidazol-2-yl)thio)methyl)-1-benzyl-1H-1,2,3-triazole (**4d**)

This compound was obtained as colorless crystals in 93% yield, mp: 239–240 °C. IR (*υ*, cm^−1^): 1505 (C=C), 1630 (C=N), 2960 (CH- Al), 3090 (CH-Ar). ^1^H NMR (400 MHz, DMSO-*d_6_*): δ_H_ = 4.44 (s, 2H, SC**H**_2_), 5.58 (s, 2H, C**H**_2_), 7.15-7.21 (m, 5H, Ar-**H**), 7.23-7.45 (m, 15H, Ar-**H**), 8.05 (s, 1H, C**H**-1,2,3-triazole) ppm. ^13^C NMR (100 MHz, DMSO-*d_6_*): δ_C_ = 27.85 (S**C**H_2_); 53.22 (N**C**H_2_); 124.04, 126.71, 127.02, 128.46, 128.61, 128.67, 128.87, 129.02, 129.22, 129.40, 129.57, 130.44, 131.14, 131.65, 134.54, 135.64, 136.45, 137.95, 142.29, 143.93 (Ar-**C**, **C**=N). Calculated for C_31_H_25_N_5_S: C, 74.52; H, 5.04; N, 14.02. Found: C, 74.88; H, 5.12; N, 14.24.

##### Ethyl-2-(4-(((1,4,5-triphenyl-1H-imidazol-2-yl)thio)methyl)-1H-1,2,3-triazol-1-yl)acetate (**4e**)

This compound was obtained as a yellow solid in 90% yield, mp: 103–105 °C. IR (*υ*, cm^−1^): 1255 (C-O), 1509 (C=C), 1610 (C=N), 1725 (C=O), 2889, 2967, (CH-Al), 3088 (CH-Ar). ^1^H NMR (400 MHz, DMSO-*d_6_*): δ_H_ 1.20 (t, 3H, *J* = 4.0 Hz, C**H**_3_), 4.14-4.19 (q, 2H, OC**H**_2_CH_3_), 4.50 (s, 2H, SC**H_2_**), 5.39 (s, 2H, NC**H**_2_), 7.19-7.27 (m, 10H, Ar-**H**), 7.36-7.48 (m, 5H, Ar-**H**), 8.04 (s, 1H, C**H**-1,2,3-triazole) ppm. ^13^C NMR (100 MHz, DMSO-*d_6_*): δ_C_ 14.43 (**C**H_3_); 27.88 (S**C**H_2_); 50.54 (N**C**H_2_), 61.34 (O**C**H_2_), 125.40, 124.43, 126.63, 126.79, 127.03, 128.63, 128.66, 128.88, 129.02, 129.45, 129.61, 130.36, 130.44, 131.15, 131.62, 131.69, 134.52, 134.58, 135.56, 135.63, 137.96, 142.35, 142.41 (Ar-**C**, **C**=N); 167.74 (**C**=O) ppm. Calculated for C_28_H_25_N_5_O_2_S: C, 67.86; H, 5.08; N, 14.13. Found: C, 67.58; H, 5.18; N, 14.39.

##### 2-(4-(((1,4,5-Triphenyl-1H-imidazol-2-yl)thio)methyl)-1H-1,2,3-triazol-1-yl)-1-(4-methoxyphenyl) ethanone (**4f**)

This compound was obtained as a yellow solid in 87% yield, mp: 154–155 °C. IR (*υ*, cm^−1^): 1560 (C=C), 1620 (C=N), 1730 (C=O), 2960 (CH-Al), 3030 (CH-Ar). ^1^H NMR (400 MHz, DMSO-*d_6_*): δ_H_ = 3.80 (s, 3H, OC**H_3_**), 4.52 (s, 2H, SC**H_2_**), 6.02 (d, 2H, NC**H_2_**), 7.16-7.29 (m, 10H, Ar-**H**), 7.32-7.48 (m, 5H, Ar-**H**), 8.06-8.20 (m, 4H, Ar-**H**), 8.10 (s, 1H, C**H**-1,2,3-triazole) ppm. ^13^C NMR (100 MHz, DMSO-*d_6_*): δ_C_ = 27.80 (S**C**H_2_); 52.42 (N**C**H_2_); 56.26 (O**C**H_3_); 118.87, 122.60, 126.09, 126.62, 128.81, 128.48, 129.65, 131.01, 131.41, 131.83, 134.44, 135.24, 136.24, 137.60, 138.45, 140.23, 141.86, 145.11, 145.29 (Ar-**C**, **C**=N); 191.13 (**C**O). Calculated for C_33_H_27_N_5_O_2_S: C, 71.07; H, 4.88; N, 12.56. Found: C, 71.32; H, 4.729; N, 12.78.

##### *N*-(3,4-Dichlorophenyl)-2-(4-(((1,4,5-triphenyl-1H-imidazol-2-yl)thio)methyl)-1H-1,2,3-triazol-1-yl)acetamide (**4g**)

This compound was obtained as a yellow solid in 90% yield, mp: 161–162 °C. IR (*υ*, cm^−1^): 1565 (C=C), 1628 (C=N), 1745 (C=O), 2899 (CH- Al), 3055 (CH-Ar), 3355 (NH). ^1^H NMR (400 MHz, DMSO-*d_6_*): δ_H_ = 4.50 (s, 2H, SC**H**_2_), 5.36 (s, 2H, NC**H**_2_), 7.11-7.64 (m, 17H, Ar-**H**), 7.96 (s, 1H, Ar-**H)**, 8.06 (s, 1H, C**H**-1,2,3-triazole), 10.82 (s, 1H, N**H**) ppm. ^13^C NMR (100 MHz, DMSO-*d_6_*): δ_C_ = 27.50 (S**C**H_2_); 52.68 (N**C**H_2_); 119.65, 120.87, 125.73, 126.95, 127.67, 128.23, 128.60, 128.99, 129.60, 130.19, 131.05, 131.38, 131.76, 134.45, 135.53, 138.92 (Ar-**C**, **C**=N); 165.11 (**C**=O). Calculated for C_32_H_24_Cl_2_N_6_OS: C, 62.85; H, 3.96; N, 13.74. Found: C, 62.69; H, 3.90; N, 13.93.

##### *N*-(4-Nitrophenyl)-2-(4-(((1,4,5-triphenyl-1H-imidazol-2-yl)thio)methyl)-1H-1,2,3-triazol-1-yl)acetamide(**4h**)

This compound was obtained as a yellow solid in 87% yield, mp: 144–145 °C. IR (*υ*, cm^−1^): 1565 (C=C), 1620 (C=N), 1700 (C=O), 2940 (CH-Al), 3080 (CH-Ar), 3340 (NH). ^1^H NMR (400 MHz, DMSO-*d_6_*): δ_H_ = 4.46 (s, 2H, SC**H**_2_), 5.36 (s, 2H, NC**H**_2_), 7.00-7.49 (m, 15H, Ar-**H**), 7.59-8.00 (m, 4H, Ar-**H**), 8.15 (s, 1H, C**H**-1,2,3-triazole), 10.60 (s, 1H, N**H**) ppm. ^13^C NMR (100 MHz, DMSO-*d_6_*): δ_C_ = 27.53 (S**C**H_2_); 52.57 (N**C**H_2_); 115.91, 116.13, 121.38, 121.36, 125.69, 127.28, 128.49, 128.92, 129.06, 129.61, 129.71, 129.99, 131.03, 134.14, 135.22, 135.64 (Ar-**C**, **C**=N); 164.53 (**C**=O). Calculated for C_32_H_25_N_7_O_3_S: C, 65.40; H, 4.29; N, 16.68. Found: C, 65.66; H, 4.33; N, 16.82.

##### *N*-(2-Fluorophenyl)-2-(4-(((1,4,5-triphenyl-1H-imidazol-2-yl)thio)methyl)-1H-1,2,3-triazol-1-yl) acetamide (**4i**)

This compound was obtained as a pale yellow solid in 89% yield, mp: 118–119 °C. IR (*υ*, cm^−1^): 1555 (C=C), 1630 (C=N), 1693 (C=O), 3070 (CH-Ar), 3330 (NH). ^1^H NMR (400 MHz, DMSO-*d_6_*): δ_H_ = 4.58 (s, 2H, SC**H**_2_), 5.39 (s, 2H, NC**H**_2_), 7.18-7.29 (m, 10H, Ar-**H**), 7.34-7.48 (m, 5H, Ar-**H**), 8.07-8.22 (m, 4H, Ar-**H**), 8.09 (s, 1H, C**H**-1,2,3-triazole), 10.85 (s, 1H, N**H**) ppm. ^13^C NMR (100 MHz, DMSO-*d_6_*): δ_C_ = 27.59 (S**C**H_2_); 52.76 (N**C**H_2_); 119.89, 120.77, 125.69, 126.95, 127.70, 128.56, 128.47, 128.80, 129.61, 131.12, 131.21, 132.07, 132.56, 133.98, 135.76, 140.57 (Ar-**C**, **C**=N); 165.23 (**C**=O). Calculated for C_32_H_25_FN_6_OS: C, 68.55; H, 4.49; N, 14.99. Found: C, 68.70; H, 4.60; N, 14.82.

##### *N*-(4-Fluorophenyl)-2-(4-(((1,4,5-triphenyl-1H-imidazol-2-yl)thio)methyl)-1H-1,2,3-triazol-1-yl) acetamide (**4j**)

This compound was obtained as a pale yellow solid in 86% yield, mp: 126–127 °C. IR (*υ*, cm^−1^): 1570 (C=C), 1625 (C=N), 1695 (C=O), 2925 (CH-Al), 3085 (CH-Ar), 3310 (NH). ^1^H NMR (400 MHz, DMSO-*d_6_*): δ_H_ = 4.51 (s, 2H, SC**H**_2_), 5.41 (s, 2H, NC**H**_2_), 7.16-7.28 (m, 10H, Ar-**H**), 7.31-7.45 (m, 5H, Ar-**H**), 8.02-8.15 (m, 4H, Ar-**H**), 8.14 (s, 1H, C**H**-1,2,3-triazole), 10.89 (s, 1H, N**H**) ppm. ^13^C NMR (100 MHz, DMSO-*d_6_*): δ_C_ = 27.59 (S**C**H_2_); 52.76 (N**C**H_2_); 119.60, 120.82, 125.85, 126.91, 127.76, 128.45, 128.54, 128.87, 129.68, 130.34, 131.11, 131.29, 131.67, 134.51, 135.48, 140.36 (Ar-**C**, **C**=N); 165.47 (**C**=O). Calculated for C_32_H_25_FN_6_OS: C, 68.55; H, 4.49; N, 14.99. Found: C, 68.78; H, 4.58; N, 14.76.

##### 4-(2-(4-(((1,4,5-Triphenyl-1H-imidazol-2-yl)thio)methyl)-1H-1,2,3-triazol-1-yl)acetamido)benzoic acid (**4k**)

This compound was obtained as a yellow solid in 86% yield, mp: 129–130 °C. IR (*υ*, cm^−1^): 1550 (C=C), 1615 (C=N), 1725 (C=O), 2510-3300 (OH, NH). ^1^H NMR (400 MHz, DMSO-*d_6_*): δ_H_ = 4.50 (s, 2H, SC**H**_2_), 5.39 (s, 2H, NC**H**_2_), 7.22-7.30 (m, 10H, Ar-**H**), 7.38-7.50 (m, 5H, Ar-**H**), 7.94 (d, 2H, *J* = 4.0 Hz, Ar-**H**), 8.14 (d, 2H, *J* = 4.0 Hz, Ar-**H**), 8.02 (s, 1H, C**H**-1,2,3-triazole), 10.78 (s, 1H, N**H**), 12.45 (s, 1H, COO**H**) ppm. ^13^C NMR (100 MHz, DMSO-*d_6_*): δ_C_ = 27.60 (S**C**H_2_); 52.67 (N**C**H_2_); 119.76, 120.81, 125.72, 126.85, 127.76, 128.38, 128.55, 128.78, 129.54, 130.54, 131.21, 131.44, 132.09, 134.56, 135.78, 141.57 (Ar-**C**, **C**=N); 167.78 (**C**=O); 168.44 (**C**=O). Calculated for C_33_H_26_N_6_O_3_S: C, 67.56; H, 4.47; N, 14.33. Found: C, 67.83; H, 4.33; N, 14.78. 

##### 1-(3,4-Dichlorophenyl)-4-(((1,4,5-triphenyl-1H-imidazol-2-yl)thio)methyl)-1H-1,2,3-triazole (**6a**)

This compound was obtained as a pale yellow solid in 85% yield, mp: 124–125 °C. IR (*υ*, cm^−1^): 1525 (C=C), 1625 (C=N), 2925 (CH- Al), 3075 (CH-Ar). ^1^H NMR (400 MHz, DMSO-*d_6_*): δ_H_ = 4.54 (s, 2H, SC**H_2_**), 7.17-7.27 (m, 10H, Ar-**H**), 7.35-7.46 (m, 5H, Ar-**H**), 7.84-7.94 (m, 2H, Ar-**H**), 8.23 (s, 1H, Ar-**H**), 8.84 (s, 1H, C**H**-1,2,3-triazole) ppm. ^13^C NMR (100 MHz, DMSO-*d_6_*): δ_C_ = 27.55 (S**C**H_2_); 120.33, 122.11, 122.68, 126.70, 127.05, 128.63, 128.97, 129.73, 131.13, 131.30, 132.29, 132.83, 141.86, 144.93 (Ar-C, C=N). Calculated for C_30_H_21_Cl_2_N_5_S: C, 64.98; H, 3.82; N, 12.63. Found: C, 64.63; H, 3.69; N, 12.88. 

##### 1-(4-Nitrophenyl)-4-(((1,4,5-triphenyl-1H-imidazol-2-yl)thio)methyl)-1H-1,2,3-triazole (**6b**)

This compound was obtained as a colorless solid in 86% yield, mp: 129–130 °C. IR (*υ*, cm^−1^): 1540 (C=C), 1605 (C=N), 2930 (CH-Al), 3085 (CH-Ar). ^1^H NMR (400 MHz, DMSO-*d_6_*): δ_H_ = 4.51 (s, 2H, SC**H_2_**), 7.13-7.25 (m, 10H, Ar-**H**), 7.38-7.50 (m, 5H, Ar-**H**), 7.90 (d, 2H, *J* = 4.0 Hz, Ar-**H**), 8.04 (d, 2H, *J* = 4.0 Hz, Ar-**H**), 8.80 (s, 1H, C**H**-1,2,3-triazole) ppm. ^13^C NMR (100 MHz, DMSO-*d_6_*): δ_C_ = 27.68 (S**C**H_2_); 119.21, 121.35, 126.45, 127.21, 128.42, 127.82, 129.52, 130.36, 131.45, 131.28, 132.41, 132.77, 134.62, 135.67, 136.31, 138.14, 141.90, 144.76 (Ar-**C**, **C**=N). Calculated for C_30_H_22_N_6_O_2_S: C, 67.91; H, 4.18; N, 15.84. Found: C, 67.66; H, 4.25; N, 15.72.

##### 1-(4-(4-(((1,4,5-Triphenyl-1H-imidazol-2-yl)thio)methyl)-1H-1,2,3-triazol-1-yl)phenyl)ethan-1-one (**6c**)

This compound was obtained as a yellow solid in 85% yield, mp: 161–163 °C. IR (*υ*, cm^−1^): 1545 (C=C), 1635 (C=N), 1765 (C=O), 2992 (CH-Al), 3079 (CH-Ar). ^1^H NMR (400 MHz, DMSO-*d_6_*): δ_H_ = 2.64 (s, 3H, C**H_3_**), 4.56 (s, 2H, SC**H_2_**), 7.18-7.28 (m, 10H, Ar-**H**), 7.36-7.47 (m, 5H, Ar-**H**), 8.07 (d, 2H, *J =* 12 Hz, Ar-**H**), 8.17 (d, 2H, *J =* 8 Hz, Ar-**H**), 8.88 (s, 1H, C**H**-1,2,3-triazole) ppm. ^13^C NMR (100 MHz, DMSO-*d_6_*): δ_C_ = 27.16 (S**C**H_2_); 27.68 (**C**H_3_); 119.62, 122.63, 126.05, 126.68, 128.80, 129.42, 130.43, 131.48, 131.80, 134.49, 135.20, 136.21, 137.57, 137.96, 139.94, 141.92, 145.02, 145.36 (Ar-**C**, **C**=N); 197.45 (**C**=O). Calculated for C_32_H_25_N_5_OS: C, 72.84; H, 4.78; N, 13.27. Found: C, 72.59; H, 4.90; N, 13.46.

##### Ethyl 4-(4-(((1,4,5-triphenyl-1H-imidazol-2-yl)thio)methyl)-1H-1,2,3-triazol-1-yl)benzoate (**6d**)

This compound was obtained as a pale yellow solid in 86% yield, mp: 134–135 °C. IR (*υ*, cm^−1^): 1210 (C-O), 1565 (C=C), 1610 (C=N), 1740 (C=O), 2975 (CH-Al), 3055 (CH-Ar). ^1^H NMR (400 MHz, DMSO-*d_6_*): δ_H_ = 1.30 (t, 3H, *J* = 4.0 Hz, C**H**_3_), 4.27-4.34 (q, 2H, OC**H**_2_), 4.52 (s, 2H, SC**H_2_**), 7.21-7.30 (m, 10H, Ar-**H**), 7.39-7.49 (m, 5H, Ar-**H**), 8.01-8.16 (m, 4H, Ar-**H**), 8.84 (s, 1H, C**H**-1,2,3-triazole) ppm. ^13^C NMR (100 MHz, DMSO-*d_6_*): δ_C_ = 14.48 (**C**H_3_); 27.49 (S**C**H_2_); 60.98 (O**C**H_2_); 119.78, 122.60, 126.51, 126.66, 128.83, 128.46, 129.47, 130.37, 131.29, 131.86, 134.52, 135.24, 136.29, 137.64, 138.11, 139.89, 141.96, 145.08, 145.42 (Ar-**C**, **C**=N); 166.22 (**C**=O). Calculated for C_33_H_27_N_5_O_2_S: C, 71.07; H, 4.88; N, 12.56. Found: C, 71.29; H, 4.99; N, 12.72.

##### 4-(4-(((1,4,5-Triphenyl-1H-imidazol-2-yl)thio)methyl)-1H-1,2,3-triazol-1-yl)benzoic acid (**6e**)

This compound was obtained as a yellow solid in 85% yield, mp: 219–220 °C. IR (*υ*, cm^−1^): 1535 (C=C), 1605 (C=N), 1720 (C=O), 2580-3350 (OH). ^1^H NMR (400 MHz, DMSO-*d_6_*): δ_H_ = 4.57 (s, 2H, SC**H_2_**), 7.15-7.30 (m, 10H, Ar-**H**), 7.39-7.48 (m, 5H, Ar-**H**), 7.92 (d, 2H, *J* = 4.0 Hz, Ar-**H**), 8.06 (d, 2H, *J* = 4.0 Hz, Ar-**H**), 8.82 (s, 1H, C**H**-1,2,3-triazole), 12.23 (s, 1H, COO**H**) ppm. ^13^C NMR (100 MHz, DMSO-*d_6_*): δ_C_ = 27.74 (S**C**H_2_); 119.26, 121.38, 126.38, 127.28, 128.53, 127.77, 129.65, 130.41, 131.37, 131.78, 132.48, 132.80, 134.78, 135.64, 136.29, 138.24, 141.82, 144.89 (Ar-**C**, **C**=N); 167.58 (**C**=O). Calculated for C_31_H_23_N_5_O_2_S: C, 70.30; H, 4.38; N, 13.22. Found: C, 70.47; H, 4.42; N, 13.34.

### 3.2. MTT Assay

The cytotoxic activity was assessed using the 3-(4,5-dimethylthiazol-2-yl)-2,5-diphenyl tetrazolium bromide (MTT) colorimetric assay as reported previously [53,54,55]. In brief, the tumor cell lines were suspended in the medium at a concentration of 1 × 104 cell/well in Corning^®^ 96-well tissue culture plates and then incubated for 24 h. The tested compounds with concentrations were then added into 96-well plates to achieve different concentrations for each compound. After incubating for 24 h, the number of viable cells was determined by the MTT test. Briefly, the medium was removed and the MTT solution (50 μL, 0.5 mg/mL in RPMI 1640 without phenol red) was added. After 3 h of incubation, the MTT solution was removed and the acquired formazan was dissolved in an isopropanol:HCl system. The 96-well plates were then incubated at 37 °C and 5% CO_2_ for 4 h. Finally, the optical density was measured at 570 nm with the microplate reader to determine the number of viable cells and the percentage of viability were calculated as [1-(ODt/ODc)] × 100% where ODt is the mean optical density of wells treated with the tested sample and ODc is the mean optical density of the untreated cells. The experiment was conducted in three independent iterations with four technical repetitions. Tests were conducted at concentrations of tested compounds ranging from 0.01 to 0.8 mM solutions. The 50% inhibitory concentration (IC_50_), the concentration required to cause toxic effects in 50% of intact cells, was estimated from graphic plots of the dose response curve for each concentration using CalcuSyn.

### 3.3. In Silico Analysis

Several physicochemical, pharmacokinetic, and pharmacodynamic attributes (i.e., topological polar surface area (TPSA), lipophilicity, absorption, distribution, metabolism, excretion and toxicity (ADMET)), molar refractivity, fragment-based drug-likeness and violations, etc., of the newly synthesized compounds were predicted through an in silico strategy utilizing the Swiss ADME tool from the Swiss Institute of Bioinformatics (http://www.sib.swiss) and ADMET SAR server (http://lmmd.ecust.edu.cn).

### 3.4. Molecular Docking Studies

The newly potent synthesized compound **4k** was docked into the active site of 3D crystallographic structure of the GSK-3β (PDB: 1UV5), used as the target model. The AutoDock 3.0 [56] and the MOE software [57] were used for all the docking calculations and molecular representations. The AutoDock Tools package was employed to generate the docking input files and analyze the docking results. A grid box size of 90 × 90 × 90 points with a spacing of 0.375 Å between the grid points was generated that covered almost the entire protein surface. The ligand was built by means of the MOE builder interface, their geometries were optimized with the CHARMm forcefield and then prepared for docking calculations with the python scripts available in the AutoDock package. Fifty runs were performed, and the resulting poses were clustered with 1.8 Å tolerance. Lamarckian GA was used for the conformational space search with the initial population set to 150, and fitness function evaluations set to 2,500,0000. The most abundant low energy clusters were selected for the analysis. The protein-ligand interaction plots were generated, using MOE 2012.10.

## 4. Conclusions

An efficient copper(I)catalyzed click reaction of imidazole-based alkyne 2 with a variety of organoazides furnished a focused library of imidazole-1,2,3-triazole hybrids carrying different un/functionalized alkyl/aryl side chains **4a**–**k** and **6a**–**e**. The targeted click products were fully characterized using different spectroscopic techniques such as IR, ^1^H, ^13^C NMR, and elemental analyses. The resulted adducts were investigated for their anticancer activity against four cancer cell lines (Caco-2, HCT-116, HeLa, and MCF-7) by the MTT assay and showed moderate to significant activity. Of these compounds, **4k** displayed potent cytotoxic activity against the cancer cell lines, especially MCF-7 with IC_50_ 0.38 µM. The study of pharmacokinetic parameters of the synthesized compounds revealed that most of the compounds are CYP2C19 and CYP3A4 inhibitors**.** All the synthesized compounds emerged to be non-carcinogenic in nature. The in silico molecular docking study was also performed recruiting GSK-3β as the promising cancer target receptor. The results of in silico studies were in accordance with the in vitro results. The overall results of this study showed that a fascinating protocol for the production of new anticancer agents targeting the GSK3β enzyme for the treatment of breast cancer was the molecular hybridization of 1,2,3-triazole to biologically interesting imidazole scaffolds with diverse aromatic substituents.

## Data Availability

No data Available.

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
