# Peer review of "Design and Synthesis of Novel Imidazole Derivatives Possessing Triazole Pharmacophore with Potent Anticancer Activity, and In Silico ADMET with GSK-3β Molecular Docking Investigations"

_ijms, 2021, doi:10.3390/ijms22031162_

Round 1

Reviewer 1 Report

This study synthesized a panel of triazole molecules using Click chemistry. These were then tested in an MTT assay with moderate results. 

Major concerns:

  1) it is unclear why GSK-3B was chosen as a potential target. Furthermore, only in silico data was reported. 

2) No other molecules were docked with GSK-3B. 

3) It is unclear what the biological discovery the paper is and the chemistry is not novel.  

Minor concerns

1) The English word choices are a bit confusing. 

2) Too much of the text is spent discussing how the authors were backed they made the correct compound.

Author Response

Response to Reviewers’ Comments 1

Ref. No.: ijms-1040176.

Title: Design and Synthesis of Novel Imidazole Derivatives Possessing Triazole Pharmacophore with Potent Anticancer Activity, and In-silico ADMET with GSK-3β Molecular Docking Investigations

Journal: International Journal of Molecular Science (IJMS)

Dear Editor,

We would like to convey our sincere gratitude to you and respected reviewers                                              for their valuable comments to improve the manuscript. The manuscript has been revised substantially as suggested. We have tried our best to follow the reviewers’ suggestion.

The following actions were performed in the revised version of the manuscript. The corrections are highlighted in red color in this revised version of the manuscript. Here, we also have added below the answers next to the queries raised by the reviewers.

Comments of Reviewer # 1:

This study synthesized a panel of triazole molecules using Click chemistry. These were then tested in an MTT assay with moderate results. 

Major concerns:

  • it is unclear why GSK-3B was chosen as a potential target. Furthermore, only in silico data was reported. 
  • Response: GSK-3B was chosen as a potential target as mentioned in the manuscript, because it is highly expressed in most of tumors and its’ aberrant activity has been implicated in many human cancers and associated with tumor progression by stabilizing components of the beta-catenin complex. So, in silico analysis was done for prediction of the novel compounds mode of action and it was succeeded predicted from binding mode as inhibitors. In future work, we will assay the most active derivatives against of panel of kinases and especially target GSK3β one.

  • No other molecules were docked with GSK-3B. 
  • Response: We did only modeling for the most active one as representative example and it is well-discussed in the manuscript.

  • It is unclear what the biological discovery the paper is and the chemistry is not novel.  
  • Response: The study combines reported wide published click chemistry protocol for novel compounds and reports anticancer evaluation for these compounds which exhibited significant activity compared to reference drug. As, the click strategy has become powerful linking reaction, to build novel 1,2,3-triazole-containing hybrids and conjugates associated with biological targets and as leads in medicinal chemistry, owing to its high degree of dependability, complete specificity, and the bio-compatibility of the reactants. Accordingly, and as continuation of our interest to explore the gold standard of click chemistry and develop novel 1,2,3-triazoles compassing active core, we have reported in the present work the synthesis of some novel imidazole-1,2,3-triazole molecular conjugates as fascinating anticancer candidates. The novelty occurs in the construction of the triphenyl imidazole molecular scaffold bonded to terminal hydrophilic triazole with S linker, however with biological testing they showed good activity anticancer profile.

Moreover, the synthesized compounds were assessed for their physicochemical, pharmacokinetic and pharmacodynamic traits. In-silico toxicity predictions revealed that all of the compounds synthesized in this work are non-carcinogenic in nature.

Minor concerns:

  • The English word choices are a bit confusing.
  • Response: We run through the manuscript and we changed the most confusing words, as much as possible.

  • Too much of the text is spent discussing how the authors were backed they made the correct compound.
  • Response: We managed the chemistry section and making it brief.

The response of authors to the editor and reviewer comments as shown above positively improved the current manuscript. So, we have to thank the reviewers for taking care and giving time to read and comment on the manuscript. Finally, please do not hesitate to contact me for any further inquiry.

Best regards

Pr. Dr. Nadjet Rezki (nadjetrezki@yahoo.fr; nrezki@taibahu.edu.sa.edu)

Reviewer 2 Report

See attached

Author Response

Response to Reviewers’ Comments 2

Ref. No.: ijms-1040176.

Title: Design and Synthesis of Novel Imidazole Derivatives Possessing Triazole Pharmacophore with Potent Anticancer Activity, and In-silico ADMET with GSK-3β Molecular Docking Investigations

Journal: International Journal of Molecular Science (IJMS)

Dear Editor,

We would like to convey our sincere gratitude to you and respected reviewers                                              for their valuable comments to improve the manuscript. The manuscript has been revised substantially as suggested. We have tried our best to follow the reviewers’ suggestion.

The following actions were performed in the revised version of the manuscript. The corrections are highlighted in red color in this revised version of the manuscript. Here, we also have added below the answers next to the queries raised by the reviewers.

Comments of Reviewer # 2:

The manuscript by Al-blewi et al., describes the design, synthesis and study of triazole compounds, aided by In-silico ADMET and Molecular modeling for the treatment of cancer. Overall, the manuscript covers appropriate steps towards obtaining the triazoles with good potency and enhanced drug-likeness which can pave the way for further exploratory studies for novel compounds with improved outcomes. There are however significant concerns that need to be taken care before the article is considered for publication. 

  1. The evidence suggesting GSK-3β as the therapeutic target is not strong. Are there any other potential or alternate target(s)?
  • Response: Thanks for your comments; The GSK-3β is a therapeutic target for different types of cancers as stated and detailed in this study https://www.nature.com/articles/s41598-020-68713-9. As well, the modeling section including the docking showed relevant binding and mode of action very similar to the original bound ligand, so the structural similarity of our target compounds behaves as the internal GSK-3β drug and hence we focused in our study on this specific target.
  1. What is the selectivity profile of the potent compounds, e.g. 4k, as well as side effects when compared to Doxorubicin? This may well be a journal style, but why is the conclusion section placed after the method section, instead of after the discussion?
  • Response: Selectivity profile of most active compounds were appeared from the in-vitro screening against MCF-7 cell line better than the reference drug with lower side effects. We also compared our compounds with doxorubicin on the basis of toxicity profile. Doxorubicin is predicted to be AMES toxic, hence mutagenic. Some of our compounds emerged to be AMES non-toxic i.e. non-mutagenic in the toxicity predictions. Just like doxorubicin, all of our synthesized compounds were non-carcinogenic in nature.
  1. There is absolutely no difference in the potency of 4k and 6e with Caco-2 cell line (IC50 of 4.67 ± 0.11 μM and 5.22 ± 0.20 μM, respectively), when compared to Doxorubicin (5.17 ± 0.25 μM). Likewise, IC50 of 0.38 μM of 4k (MCF-7) is not significantly different from an IC50 of 0.65 μM by Doxorubicin. In-silico ADMET is clearly a useful way to predict physicochemical properties, pharmacokinetics, drug- likeness and medicinal chemistry friendliness of compounds and helps narrow down potential compounds for synthesis and subsequent biochemical studies. It is therefore, not clear why this technique was not used prior to selecting compounds for synthesis instead of after the functional study.
  • Response: We appreciate the reviewer comments and we amended the suggested changes in the manuscript. The protocol of synthesis should be done for compounds series with various substituents to show which one will be potent and put SAR screen, then the last step will be the computational analysis of the actives not all. In-silico ADMET studies were performed to get an insight on the physicochemical, pharmacodynamic and pharmacokinetic traits of the synthesized compound and to integrate in-vitro and in-silico   
  1. The manuscript has several grammatical errors and badly worded sentence constructions and require careful editing especially the molecular modeling study section, but also throughout the paper. A few examples are a. It was identified that Glycogen synthase kinase-3 beta (GSK-3β) as promising target an overexpressed oncogene in different breast cancers, we further evaluated the binding modes of our potent compound 4k as selective MCF-7 cytotoxic agent against GSK-3β ATP binding pocket and confirmed the overall prognostic dependency of breast cancer patients on GSK-3β inhibition effect. - badly constructed sentence and needs revision b. The forecasted physicochemical characteristics are in agreement with the exercised criteria and are presumed to possess good bioavailability score as all the compounds have TPSA ≤ 140 Å…. (What is meant by exercise criteria???) In-silico forecast of physicochemical descriptors, drug likeness or ADME (adsorption, distribution, metabolism and excretion) have escalated the possibility of detecting new lead compounds in a much shorter time span as compared to the conventional procedure – what conventional procedure???. Also reword the sentence to make clear d. These results suggested that, the designed triazoles are promising candidates for the future anticancer molecule discovery protocols were expectable. – sentence not clear e. ……was found to be the most potent derivative against …… f. Not clear what they meant by this statement “Interestingly, a preliminary structure– activity relationship analysis urged that the derivatives 4e, 6c and 6d have an ester and/or acetyl functionality in their structures.” REVIEW g. …synthesized triazoles flaunted (replace flaunted with showed) excellent to good activity h. ….and to optimize screening and trials by gazing at only the promising drug candidates. What is meant by gazing??? i. drug-like’ traits that can be thrived into structure-property association to complement target specified structure-activity alliance. – sentence not clear j. … with the glycogen synthase kinase-3 (GSK-3). GSK-3 ……
  2. a) Corrected in the manuscript
  3. b) The exercised criteria are Lipinki, Ghose, Veber, Egan and Muegge rule for drug likeness.
  4. c) The conventional procedure includes random screening of chemicals found in nature or blindly synthesized in laboratories. They are time-consuming, and it requires substantial investments in terms of capital, human resources, research skill, and technological expertise. For example an NCE targeting the peroxisome proliferator activated receptor γ family ragaglitazar (NN622), a dual acting insulin sensitizer, was found to be positive in the carcinogenic bioassay study (urinary bladder tumors were observed in mice and rats), and the drug had to be dropped during phase III trials, after many millions of dollars had been spent on its development. The problems with this traditional method are long design cycle and high cost. Modern approach including structure-based drug design with the help of informatic technologies and computational methods has speeded up the drug discovery process in an efficient manner. An improved generation of softwares with easy operation and superior computational tools to generate chemically stable and worthy compounds with refinement capability has been developed. These tools can tap into cheminformation to shorten the cycle of drug discovery, and thus make drug discovery more cost-effective.
  5. d) Corrected in the manuscript.
  6. e) Corrected in the manuscript.
  7. f) Corrected in the manuscript.
  8. g) Corrected in the manuscript.
  9. h) Gazing in general means look steadily and intently, especially in admiration, surprise, or thought. So we intently looked and thoughted about promising candidates only.
  10. i) Corrected in the manuscript

The response of authors to the editor and reviewer comments as shown above positively improved the current manuscript. So, we have to thank the reviewers for taking care and giving time to read and comment on the manuscript. Finally, please do not hesitate to contact me for any further inquiry.

Best regards

Pr. Dr. Nadjet Rezki (nadjetrezki@yahoo.fr; nrezki@taibahu.edu.sa.edu)

Reviewer 3 Report

The manuscript “Design and Synthesis of Novel Imidazole Derivatives Possessing Triazole Pharmacophore with Potent Anticancer Activity, and In-silico ADMET with GSK-3β Molecular Docking Investigations” by Fawzia Al-blewi et al. is an interesting combined experimental and theoretical report on new structures which can be possibly followed and optimized with regard to their bioactivity properties. The manuscript is sound scientifically, but some procedures are described in insufficient detail to ensure full possibility to reproduce the results. I gladly recommend its publication when the following minor issues are resolved.

1. Please check carefully the experimental methods. There are some inconsistencies there. Line 264: the vendor for the chemicals is not provided. Line 278: the “12 mmol” corresponds to the imidazole or the sodium methoxide? Line 294: composition of the eluent could be given (e.g. hexane – ethyl acetate 1:1). Lines 460-461: web links to the ADMET tool are missing. Line 474: are the quantum-mechanical calculations used at all in the text? What is their role?

2. Line 56: “emanate “ should be “emanated”

3. Figure 1, bottom left: should “cytoxic” be “cytotoxic”? Please correct.

4. Line 152: the fragment “urged that the derivatives 4e, 6c and 6d, these was probably due to the presence…” is unclear, please revise.

5. Results of Table 2: please comment on the fact that the compounds are predicted to be “insoluble”. Does it hamper their bioavailability?

6. Section 2.2.3 on docking: please include the information on the residues interacting with the 6-bromoindirubin-3'-oxime. There might be less of them, and this would explain better binding of the 4K.

7. Line 257: Please revise “The results of in-silico results”.

End of reviewer remarks.

Author Response

Response to Reviewers’ Comments 3

Ref. No.: ijms-1040176.

Title: Design and Synthesis of Novel Imidazole Derivatives Possessing Triazole Pharmacophore with Potent Anticancer Activity, and In-silico ADMET with GSK-3β Molecular Docking Investigations

Journal: International Journal of Molecular Science (IJMS)

Dear Editor,

We would like to convey our sincere gratitude to you and respected reviewers                                              for their valuable comments to improve the manuscript. The manuscript has been revised substantially as suggested. We have tried our best to follow the reviewers’ suggestion.

The following actions were performed in the revised version of the manuscript. The corrections are highlighted in red color in this revised version of the manuscript. Here, we also have added below the answers next to the queries raised by the reviewers.

Comments of Reviewer # 3:

The manuscript “Design and Synthesis of Novel Imidazole Derivatives Possessing Triazole Pharmacophore with Potent Anticancer Activity, and In-silico ADMET with GSK-3β Molecular Docking Investigations” by Fawzia Al-blewi et al. is an interesting combined experimental and theoretical report on new structures which can be possibly followed and optimized with regard to their bioactivity properties. The manuscript is sound scientifically, but some procedures are described in insufficient detail to ensure full possibility to reproduce the results. I gladly recommend its publication when the following minor issues are resolved.

Please check carefully the experimental methods. There are some inconsistencies there.

  • Response: The experimental methods were checked as requested (See manuscript).

Line 264: the vendor for the chemicals is not provided.

  • Response: Provided as requested (See manuscript).

Line 278: the “12 mmol” corresponds to the imidazole or the sodium methoxide?

  • Response: Both of imidazole and sodium methoxide as it is equivalent, it was corrected (See manuscript).

 Line 294: composition of the eluent could be given (e.g. hexane – ethyl acetate 1:1).

  • Response: Composition of the eluent was given (See manuscript).

Lines 460-461: web links to the ADMET tool are missing.

  • Response: We appreciate the suggestions made by the reviewer and the web links to the ADMET tools were added to the manuscript (See manuscript).

Line 474: are the quantum-mechanical calculations used at all in the text? What is their role?

  • Response: The quantum-mechanical calculations are NOT used at all in the work, and section was put by mistake but was deleted from methodology (See manuscript).

Figure 1, bottom left: should “cytoxic” be “cytotoxic”? Please correct (See manuscript).

  • Response: Corrected as requested (See manuscript).

Line 56: “emanate “ should be “emanated”

  • Response:

Figure 1, bottom left: should “cytoxic” be “cytotoxic”? Please correct.

  • Response: Corrected as requested (See manuscript).

Line 152: the fragment “urged that the derivatives 4e, 6c and 6d, these was probably due to the presence…” is unclear, please revise.

  • Response: It was revised and corrected (See manuscript).

Results of Table 2: please comment on the fact that the compounds are predicted to be “insoluble”. Does it hamper their bioavailability?

  • Response: The oral bioavailability depends on several factors including aqueous solubility, drug permeability, dissolution rate, first-pass metabolism, presystemic metabolism, and susceptibility to efflux mechanisms. The most frequent causes of low oral bioavailability are attributed to poor solubility and low permeability. Yes, if the compounds are insoluble in water, it will affect their bioavailability (See manuscript).

Section 2.2.3 on docking: please include the information on the residues interacting with the 6-bromoindirubin-3'-oxime. There might be less of them, and this would explain better binding of the 4K.

  • Response: All information about the interacting residues with the bound ligand is presented in detail in manuscript (See manuscript).

Line 257: Please revise “The results of in-silico results”.

  • Response: We appreciate the reviewer comments and we amended the suggested revision in the manuscript (See manuscript).

The response of authors to the editor and reviewer comments as shown above positively improved the current manuscript. So, we have to thank the reviewers for taking care and giving time to read and comment on the manuscript. Finally, please do not hesitate to contact me for any further inquiry.

Best regards

Pr. Dr. Nadjet Rezki

E-mail: nadjetrezki@yahoo.fr; nrezki@taibahu.edu.sa.edu

Reviewer 4 Report

The submitted manuscript entitled “Design and synthesis of novel imidazole derivatives possessing triazole pharmacophore with potent anticancer activity, and in-silico ADMET with GSK-3β molecular docking investigations” by Fawzia Al-blewi et al. refers to the synthesis and the structure-based investigation of new series of novel imidazole-1,2,3-triazole hybrids. On the whole, the provided paper is well-written and seems to be appealing for the scientific community; however some minor concerns prevent the manuscript from suggesting for publication in the present form.

From my perspective some points should be addressed before publishing the manuscript in IJMS.

Let me introduce some major issues.

  1. In section 2.2.2 the physicochemical variables for the selected compounds are calculated. In the multidimensional space the similarity analysis is advisable using the chosen projection procedures, for instance, principal component analysis (PCA) or hierarchical clustering (HCA).
  2. The final sentence in the conclusive part is not clear for me. What does ‘molecular hybridization of 1,2,3 triazole to biological interesting imidazole scaffolds with different aromatic tails’ mean?

There is also a number or minor issues:

  1. Line 86. It should read: Figure 1. The workflow of the study.
  2. Lines 94-96,98. The font size should be corrected.
  3. Line 118. Dot is missing.
  4. Line 113. Space should be removed.
  5. Line 149. I would avoid the emotional words ‘excellent’.
  6. Line 177. Surface is not expressed in A, but in A2.
  7. Line 203. It should read: Figure 2.
  8. Line 241. It should read: Figure 3. Docking pose
  9. The references should be unified according to the journal requirements (for instance, refs. 20, 33, 51, etc.
  10. Why refs 34-38 are divided into sub-references a and b?
  11. In ref. 49 page format and price are provided. Is it necessary?

Author Response

Response to Reviewers’ Comments 4

Ref. No.: ijms-1040176.

Title: Design and Synthesis of Novel Imidazole Derivatives Possessing Triazole Pharmacophore with Potent Anticancer Activity, and In-silico ADMET with GSK-3β Molecular Docking Investigations

Journal: International Journal of Molecular Science (IJMS)

Dear Editor,

We would like to convey our sincere gratitude to you and respected reviewers                                              for their valuable comments to improve the manuscript. The manuscript has been revised substantially as suggested. We have tried our best to follow the reviewers’ suggestion.

The following actions were performed in the revised version of the manuscript. The corrections are highlighted in red color in this revised version of the manuscript. Here, we also have added below the answers next to the queries raised by the reviewers.

Comments of Reviewer # 4:

The submitted manuscript entitled “Design and synthesis of novel imidazole derivatives possessing triazole pharmacophore with potent anticancer activity, and in-silico ADMET with GSK-3β molecular docking investigations” by Fawzia Al-blewi et al. refers to the synthesis and the structure-based investigation of new series of novel imidazole-1,2,3-triazole hybrids. On the whole, the provided paper is well-written and seems to be appealing for the scientific community; however some minor concerns prevent the manuscript from suggesting for publication in the present form.

From my perspective some points should be addressed before publishing the manuscript in IJMS.

Some major issues.

In section 2.2.2 the physicochemical variables for the selected compounds are calculated. In the multidimensional space the similarity analysis is advisable using the chosen projection procedures, for instance, principal component analysis (PCA) or hierarchical clustering (HCA).

  • Response: We appreciate the reviewer comment, but these analyses were not performed (See manuscript).

The final sentence in the conclusive part is not clear for me. What does ‘molecular hybridization of 1,2,3 triazole to biological interesting imidazole scaffolds with different aromatic tails’ mean?

  • Response: The sentence was rephrased to be clear (See manuscript).

There is also a number or minor issues:

Line 86. It should read: Figure 1. The workflow of the study.

  • Response: The Figure 1 was managed to be read in easy way and modified in manuscript.

Lines 94-96, 98. The font size should be corrected.

  • Response: The font size was corrected (See manuscript).

Line 118. Dot is missing.

  • Response: Dot was inserted (See manuscript).

Line 113. Space should be removed.

  • Response: The space was removed (See manuscript).

Line 149. I would avoid the emotional words ‘excellent’.

  • Response: The words `excellent` were replaced by very good (See manuscript).

Line 177. Surface is not expressed in A, but in A2.

  • Response: We appreciate the reviewer comments and we amended the suggested revision in the manuscript (See manuscript).

Line 203. It should read: Figure 2.

  • Response: We appreciate the reviewer comments and we amended the suggested revision in the manuscript (See manuscript).

Line 241. It should read: Figure 3. Docking pose

  • Response: the figure is managed in manuscript (See manuscript).

The references should be unified according to the journal requirements (for instance, refs. 20, 33, 51, etc. Why refs 34-38 are divided into sub-references a and b?

In ref. 49 page format and price are provided. Is it necessary?

  • Response: All references were revised and unified to the journal requirements (See manuscript).

The response of authors to the editor and reviewer comments as shown above positively improved the current manuscript. So, we have to thank the reviewers for taking care and giving time to read and comment on the manuscript. Finally, please do not hesitate to contact me for any further inquiry.

Best regards

Pr. Dr. Nadjet Rezki

E-mail: nadjetrezki@yahoo.fr; nrezki@taibahu.edu.sa.edu
